# Neuroanatomy accounts for age-related changes in risk preferences

Michael A. Grubb[1,2], Agnieszka Tymula[3], Sharon Gilaie-Dotan[4,5], Paul W. Glimcher[2,6] & Ifat Levy[7]

Many decisions involve uncertainty, or 'risk', regarding potential outcomes, and substantial empirical evidence has demonstrated that human aging is associated with diminished tolerance for risky rewards. Grey matter volume in a region of right posterior parietal cortex (rPPC) is predictive of preferences for risky rewards in young adults, with less grey matter volume indicating decreased tolerance for risk. That grey matter loss in parietal regions is a part of healthy aging suggests that diminished rPPC grey matter volume may have a role in modulating risk preferences in older adults. Here we report evidence for this hypothesis and show that age-related declines in rPPC grey matter volume better account for age-related changes in risk preferences than does age per se. These results provide a basis for understanding the neural mechanisms that mediate risky choice and a glimpse into the neurodevelopmental dynamics that impact decision-making in an aging population.

[1] Department of Psychology, Trinity College, 300 Summit Street, Hartford, Connecticut 06106, USA. [2] Center for Neural Science, New York University, 4 Washington Place, Room 809, New York, New York 10003, USA. [3] School of Economics, University of Sydney, Room 370, Merewether Building (H04), Sydney, New South Wales 2006, Australia. [4] Institute of Cognitive Neuroscience, University College London, Alexandra House, 17 Queen Square, London WC1N 3AR, UK. [5] Vision Science and Optometry, Bar Ilan University, Ramat Gan 5290002, Israel. [6] Institute for the Interdisciplinary Study of Decision Making, New York University, 300 Cadman Plaza West, Suite 702, Brooklyn, New York 11201, USA. [7] Section of Comparative Medicine and Department of Neuroscience, Yale School of Medicine, PO Box 208016, New Haven, Connecticut 06520, USA. Correspondence and requests for materials should be addressed to I.L. (email: Ifat.levy@yale.edu).

n just over 30 years, adults over the age of 60 are expected to globally outnumber children for the first time in history[1]. A mechanistic understanding of how healthy human aging impacts decision-making will be critical in tackling the challenges inherent in such an unprecedented demographic shift. Most decisions involve uncertainty, or 'risk', regarding potential outcomes. Understanding age-related changes in risk preferences is, therefore, an important first step in forecasting how decisions made by an aging population might impact, for better or worse, political and economic processes at the global and local levels.

When older adults choose between certain and risky monetary rewards whose outcome probabilities are explicitly stated (that is, decisions from description), substantial empirical evidence supports the common intuition that aging is associated with increased aversion to risk[2–6]. Which neurobiological markers of aging might be associated with this change in preference? We recently identified a region in right posterior parietal cortex (rPPC) whose grey matter volume (GMV) accounts for individual variation in risk preferences in young adults, such that decreased rPPC GMV is predictive of increased risk aversion[7]. Grey matter loss is part of healthy aging, with parietal regions showing particularly enhanced local declines[8]. Therefore, an intriguing possibility is that reduced rPPC GMV, rather than age per se, may best account for the increased risk aversion empirically observed during aging. Here we tested this hypothesis in a sample of urban adults whose ages span seven decades.

Though both older age and decreased rPPC GMV are associated with risk tolerance, when the independent contributions of these factors are assessed, rPPC GMV still accounts for changes in risk tolerance, whereas age does not. These results refine and extend our existing understanding of the relationship between aging and risk tolerance by attributing behavioural changes to an age-related process (that is, changes in grey matter thickness) rather than to chronological age itself.

## Results

**Tolerance for risk**. Risk preferences were assessed using a well-validated, incentive-compatible procedure[4,9–13]. Fifty-two participants (18–88 years old, mean: 54.7, s.d.:22.1; 30 females) made 60 binary choices between a certain gain of $5 and a lottery whose monetary value and probability of payout were systematically manipulated (Fig. 1). We modelled the expected utility (EU) of each option using the functional form:

$$EU(v, p) = p \cdot v^{\alpha}$$

where $v$ = value (amount), $p$ = probability, and $\alpha$ (alpha) = the risk preference parameter, with larger alpha values indicative of increased risk tolerance (that is, risk aversion increases as alpha decreases). Choice data were fit, and alpha estimated, using maximum likelihood, with the probability of choosing the lottery

($P_{lottery}$) given by a logistic function:

$$P_{lottery} = \frac{1}{1 + e^{\left(EU_{safe} - EU_{lottery}\right)/\sigma}}$$

where $EU_{safe}$ ($EU_{lottery}$) indicates the EU of the certain (lottery) option, and $\sigma$ indicates the slope of the choice function. To account for within and between participant variabilities in an assumption-free and statistically rigorous manner, we fit choice data from all participants simultaneously, clustering the standard errors to account for participant-level correlations[7,14,15].

**rPPC grey matter volume**. Using voxel-based morphometry (VBM), we sampled GMV in the rPPC region-of-interest, which was defined independently based on an earlier study (Fig. 2a; MNI coordinates 27, − 78, 48; spatial extent, 1,232 mm[3]; from ref. 7, Study 1; mask download available at https://yale.box.com/v/levylab-gilaie-dotan-etal-2014). In Fig. 2b, rPPC GMV is plotted as a function of age and confirms that GMV in our parietal region-of-interest does indeed decrease with age in our lifespan sample (Pearson correlation, $n = 52$, $r = -0.66$, $P = 1.1–07$).

**Brain-behaviour relationships**. To assess the relationship between risk preferences and our variables of interest we allowed alpha, the risk preference parameter, to vary during the estimation procedure as a linear function of age (Model 1: $\alpha = \beta_1 \times age + \beta_0$) and rPPC GMV (Model 2: $\alpha = \beta_1 \times rPPC$

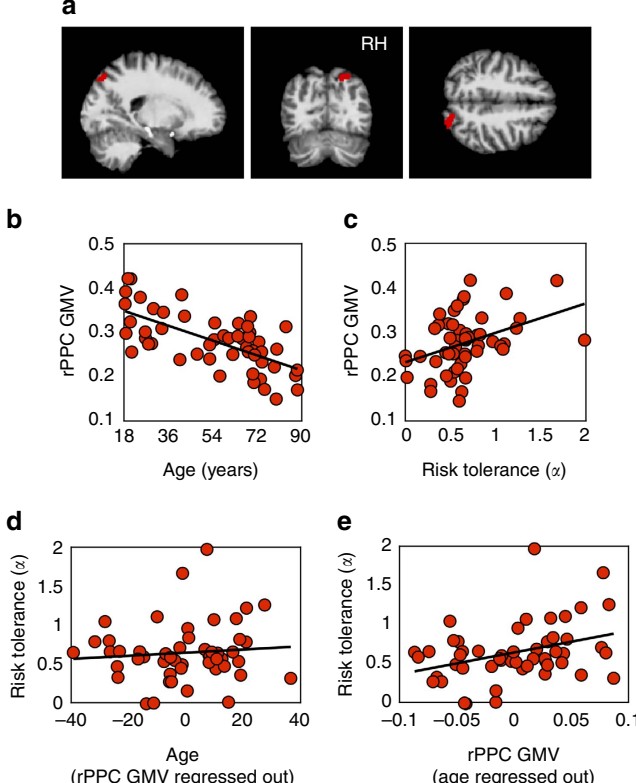

**Figure 2 | rPPC grey-matter volume accounts for risk tolerance after controlling for age.** (**a**) *A priori* defined region of interest: right posterior parietal cortex (rPPC). (**b**) rPPC grey matter volume plotted as a function of age for individual participants ($n = 52$). (**c**) rPPC grey matter volume plotted as a function of risk tolerance for individual participants. (**d**) Risk tolerance as a function of age, controlling for rPPC grey matter volume, plotted for individual participants. (**e**) Risk tolerance as a function of rPPC grey matter volume, controlling for age, plotted for individual participants.

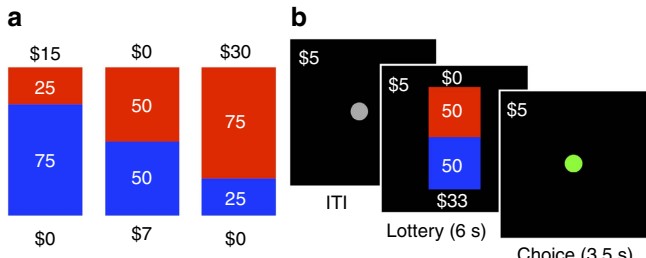

**Figure 1 | Experimental design.** (**a**) Example lotteries representing a 25, 50, 75% chance of gaining $15, $7, $30, respectively. (**b**) Example trial sequence.

**Table 1 | Estimated coefficients and Bayesian Information Criteria values for each model.**

|  | Model 1 | Model 2 | Model 3 | Model 4 | Model 5 |
|---|---|---|---|---|---|
| *Risk tolerance ($\alpha$)* |  |  |  |  |  |
| Age | − 0.003** (0.001) | — | -0.0004 (0.0017) | — | 0.001 (0.002) |
| rPPC GMV | — | 1.338*** (0.382) | 1.247* (0.585) | 0.974* (0.466) | 1.106• (0.567) |
| Global GMV | — | — | — | 0.0007 (0.0005) | 0.0009 (0.0007) |
| Constant | 0.669*** (0.064) | 0.152 (0.112) | 0.200 (0.241) | -0.162 (0.277) | -0.357 (0.517) |
|  |  |  |  |  |  |
| *Logistic slope ($\sigma$)* |  |  |  |  |  |
| Constant | 1.016*** (0.168) | 0.986*** (0.156) | 0.985*** (0.156) | 0.971*** (0.158) | 0.970*** (0.158) |
|  |  |  |  |  |  |
| *Bayesian Information Criteria* |  |  |  |  |  |
| Value | 3,084 | 3,024 | 3,031 | 3,011 | 3,016 |
| Rank | 5 | 3 | 4 | 1 | 2 |

Standard errors, s.e.'s, in parentheses; s.e.'s clustered on participant. Coefficients significantly different from zero indicated by asterisks: ***$P<0.001$; **$P=0.01$; *$P<0.05$; •$P=0.051$.

**Table 2 | Estimated coefficients for each model.**

|  | Model 1 | Model 2 | Model 3 | Model 4 | Model 5 |
|---|---|---|---|---|---|
| *Risk tolerance ($\alpha$)* |  |  |  |  |  |
| Age | − 0.003* (0.001) | — | -0.0006 (0.0019) | — | 0.0008 (0.002) |
| rPPC GMV | — | 1.37*** (0.417) | 1.252* (0.585) | 1.017* (0.474) | 1.111• (0.566) |
| Global GMV | — | — | — | 0.0008 (0.0005) | 0.0009 (0.0007) |
| Gender | 0.014 (0.070) | 0.011 (0.057) | 0.020 (0.062) | 0.032 (0.055) | 0.027 (0.058) |
| Constant | 0.700*** (0.063) | 0.137 (0.138) | 0.200 (0.241) | -0.238 (0.305) | -0.380 (0.517) |
|  |  |  |  |  |  |
| *Logistic slope ($\sigma$)* |  |  |  |  |  |
| Constant | 1.015*** (0.168) | 0.986*** (0.156) | 0.984*** (0.156) | 0.969*** (0.157) | 0.969*** (0.157) |

Standard errors, s.e.'s, in parentheses; s.e.'s clustered on participant. Coefficients significantly different from zero indicated by asterisks: ***$P<0.001$; **$P<0.01$; *$P<0.05$; •$P=0.050$.

GMV $+ \beta_0$). As predicted by previous research[4,7], we found a significant negative relationship between alpha and age (Z-test, $n=3,077$, s.e.'s clustered on 52 participants, $z=-2.58$, $P=0.01$; Table 1, Model 1) and a significant positive relationship between alpha and rPPC GMV (Z-test, $n=3,077$, s.e.'s clustered on 52 participants, $z=3.51$, $P=0.0004$; Table 1, Model 2). Controlling for gender in each model revealed no effect of gender on risk tolerance and did not qualitatively change the relationship between risk tolerance and age/rPPC grey matter (Table 2). To illustrate the positive correlation between risk tolerance and parietal grey matter, choice data were modelled at the individual level and the risk tolerance parameter derived from those fits (alpha) is plotted as a function of each individual participant's rPPC GMV in Fig. 2c.

Does the decline of rPPC GMV in fact account for the age-related increase in risk aversion? To answer this question we employed a standard econometric approach to obtain an unbiased estimate of the degree to which age-related variation in risk attitude can be attributed more parsimoniously to GMV: we allowed alpha to vary with both age and rPPC GMV (Model 3: $\alpha = \beta_1 \times \text{age} + \beta_2 \times \text{rPPC GMV} + \beta_0$) and again found a significant positive relationship between alpha and rPPC GMV (Z-test, $n=3,077$, s.e.'s clustered on 52 participants, $z=2.13$, $P=0.033$). Critically, however, when the linear regression was computed in this manner, age no longer had any influence on alpha (Z-test, $n=3,077$, s.e.'s clustered on 52 participants, $z=-0.24$, $P=0.81$), indicating that rPPC GMV, and not age per se, modulates risk preferences (Table 1, Model 3). To illustrate this effect for individual participants, we plot the independent contributions of these two factors on risk preferences: alpha as a function of age after regressing out the contribution of rPPC GMV (Fig. 2d) and as a function of rPPC GMV after regressing

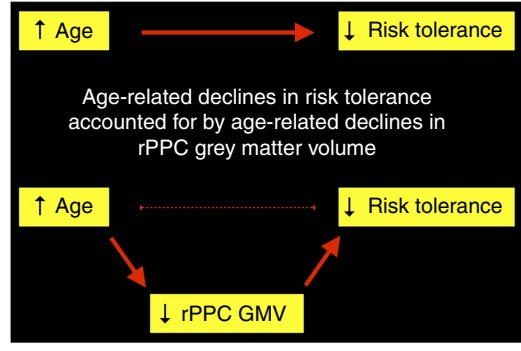

out the contribution of age (Fig. 2e). A schematic of the main results is presented in Fig. 3.

Two additional models confirmed that these results are specific to local grey matter decline in the rPPC, rather than global, age-related changes in grey matter thickness. When global GMV (Model 4: $\alpha = \beta_1 \times \text{rPPC GMV} + \beta_2 \times \text{global GMV} + \beta_0$) and global GMV + age (Model 5: $\alpha = \beta_1 \times \text{age} + \beta_2 \times \text{rPPC GMV} + \beta_3 \times \text{global GMV} + \beta_0$) were included, increased rPPC GMV still predicted increased risk tolerance (Z-tests, $n=3,077$, s.e.'s clustered on 52 participants, $z=2.09$, $P=0.037$, Model 4; $z=1.95$, $P=0.051$, Model 5), whereas neither global GMV nor age did (Table 1). Bayesian Information Criteria[16] values indicate that these final two neurobiologically comprehensive models best characterize the choice process, despite the penalties incurred for additional parameters (Table 1). Finally, to ensure that our results

**Figure 3 | Overview.** Schematic presentation of results.

do not depend on the functional form of the model, we used multiple regression to determine if individual age and rPPC GMV can predict the proportion of lottery choices that each participant made, with fewer lottery choices indicative of greater risk aversion: proportion of lottery choice $= \beta_1 \times$ age $+ \beta_2 \times$ rPPC GMV $+ \beta_0$. We found converging evidence that rPPC GMV, but not age, accounts for changes in risk preferences using this model-free approach (T-tests, $n = 52$, $\beta_1 = 0.00$, $t = 0.77$, $P = 0.78$; $\beta_2 = 1.31$, $t = 1.85$, $P = 0.035$; $p$s one-tailed in predicted directions).

While the primary aim of the current study was to test a specific hypothesis regarding the rPPC's role in modulating age-related changes in risk tolerance, we also conducted an exploratory whole-brain VBM analysis to evaluate whether GMV is predictive of risk tolerance in any additional brain regions. In a voxel-wise manner, multiple regression was used to compute the linear relationship between risk tolerance and GMV, controlling for age, gender and global GMV. No clusters showed a significant relationship between GMV and risk tolerance after the stringent corrections needed to combat false discoveries in exploratory whole-brain analyses. Given that this is the third independent data set showing a significant relationship between rPPC GMV and risk tolerance, the likelihood that we are reporting a repeated false discovery is extremely low. While we cannot definitively rule out the possibility that additional regions' structure and function contribute to age-related changes in risk tolerance, our *a priori* hypothesis-driven ROI analyses point to a clear role of the rPPC in these processes.

Choice data were collected in a magnetic resonance imaging (MRI) scanner during the acquisition of functional scans (manuscript in preparation). Although in theory the scanner environment may affect individual risk attitudes, we note that age-based estimates of risk tolerance derived from Model 1 are comparable to those obtained outside the scanner: the risk tolerance parameter (alpha) is predicted to drop slowly with each passing year, from 0.61 at 21 years of age to 0.42 by 90 years old. These estimates of risk tolerance fall within the 95% confidence intervals for age-specific alpha values reported previously by our group[4] in a task where choices were made on a desktop computer.

## Discussion

To advance a neurobiological understanding of age-related changes in decision-making, we must link changes in behaviour to neurobiological processes that unfold across the lifespan rather than to chronological age itself[17]. That the loss of GMV in rPPC better accounts for changing risk preferences than does age provides a remarkably simple explanation of this type at the level of brain macrostructure. This finding also furthers our understanding of the neurobiological basis of risk preferences at large. While the general relationship between structural MRI measures and the neural microstructure is unclear[18], in aging there are multiple changing factors that can account for GMV decline, such as changes in synaptic density, neuronal distribution size, dendritic arborization, molecular shifts and others[19–22]. These could all affect efficient neural coding and thus are compatible with computational theories suggesting that risk aversion results from limited neural computational capacity[23].

The current study only includes adults over 18 years old. As in our previous study[4], risk tolerance decreases monotonically within this age range. When adolescents are taken into account, however, the lifespan trajectory for risk tolerance may be described by a U-shaped function, with adolescents showing higher risk aversion compared with young and midlife adults[9]. This raises an intriguing hypothesis for future research—that increased risk aversion in adolescence might be parsimoniously

accounted for by changes in rPPC GMV during childhood and adolescent development.

Finally, it should be emphasized that risk taking in real life involves multiple, dissociable components[10,24–26] (for example, attitudes to 'ambiguity', or unknown risks, loss aversion, learning of implicit probability structures). Future research will need to address the relationship between changes in neuroanatomy, changes in neural patterns of activity, computational models of the decision-making process and risk-taking behaviours.

## Methods

**Participants.** Fifty-two adults (18–88 years old, mean: 54.7, s.d.: 22.1; 30 females) participated in the study. Based on the effect sizes observed in our previous studies, our sample size should provide adequate statistical power to detect significant relationships between age and risk tolerance[4], as well as rPPC risk tolerance and GMV[7]. Participants were right-handed, had normal or corrected to normal vision, were not taking medication for any psychiatric condition or developmental disorder, provided informed consent in accordance with the NYU IRB, and were recruited via bulletin boards and community centers. Three additional participants (ages: 31, 38, 48) completed the study but were excluded from the analyses presented here: these participants chose the objectively worse option (for example, some chance of $5 over $5 for sure, see Task section below) >50% of the time, which indicates a preference for less, rather than more money or a misunderstanding of the task; we could not, in principle, estimate risk preferences for these participants[4].

For participants aged 65 and older, we used the Mini Mental State Examination (Psychological Assessment Resources) to exclude overt cognitive impairment; all participants received scores between 27 and 30 (mean: 28.9, s.d.:1.06), indicating no overt cognitive impairment. Participants received $50 for taking part in the study, as well as a bonus payment (see below).

**Task.** Participants received detailed explanations of the task and of the bonus payment procedure and were required to pass task comprehension questions before completing practice trials. In the experiment itself, participants made 60 binary choices between a certain gain of $5 and a lottery whose monetary value (20 amounts: $5–$120) and probability of payout (3 levels: 0.25, 0.5, 0.75) were systematically manipulated. Each lottery was represented by an image of a bag containing 100 coloured poker chips, some red and some blue (Fig. 1); these images corresponded to physical bags that were present in the experimental room. The size of the coloured areas and the numbers written inside indicated the number of chips of each colour in the bag. Above and below each colour, a number indicated how much a chip of that colour would be worth if it were drawn from the bag. These pure risk choices were made as part of a larger study that included an additional 60 choices between a certain gain of $5 and ambiguous lotteries (20 amounts: $5–$120) whose exact probabilities were unknown (see 'Ambiguous Lotteries' below).

All choices were made in the scanner during functional MRI scans whose data are not part of this study. An additional four trials were included to accommodate functional MRI analyses. Functional data from the first trial of each run would be discarded, and for most participants (44 of 52), we started each block with a choice between the same certain gain of $5 and 50% chance of $4. The additional four trials for the remaining eight participants were lotteries of $140 in value (two risky: 25% and 50%; and two ambiguous) and were randomly intermingled with the other trials. Only the 60 pure-risk trials described above were included in the analyses presented here, and thus, the choice set used to assess risk preferences is identical across all participants. Trial order was randomized independently for each participant.

**Bonus payment.** One trial was randomly selected at the end of the experiment, and the choice made on this trial determined a participant's bonus earnings: $5 if the certain amount was chosen, $0 or some larger amount if the lottery was chosen. For lottery realizations, participants reached into a physical bag with the correct number of red and blue chips inside; the colour of the drawn chip corresponded to a value amount on each trial (Fig. 1) and determined the bonus payment.

**Individual risk preferences.** Choice data were fit, separately for individual participants, using the maximum likelihood procedure described in the main text. The slope of the logistic choice function was held constant during the estimation procedure using the estimate obtained for the corresponding population-level analysis (Fig. 2c; Table 1, Model 2; Fig. 2d,e: Table 1, Model 3).

**Structural MRI.** Anatomical images were collected using a Siemens Allegra 3T head-only scanner at the NYU Center for Brain Imaging. High-resolution T1-weighted anatomical images ($1 \times 1 \times 1$ mm$^3$) were acquired with an MPRAGE pulse sequence (TI = 900 ms, sagittal slices, $256 \times 256$ matrix).

# ARTICLE

**VBM analysis.** VBM analyses were performed with SPM12 (http://www.fil.ion.ucl.ac.uk/spm/). Using the VBM8 toolbox (http://www.neuro.uni-jena.de/vbm/), structural images were normalized to MNI stereotactic space, segmented into grey matter, white matter and cerebrospinal fluid, and spatially smoothed with a Gaussian kernel (FWHM = 8 mm). For each participant, GMV in our rPPC region-of-interest (MNI coordinates 27, − 78, 48; 1,232 mm³; ref. 7, Study 1) was sampled using the MarsBaR toolbox (http://marsbar.sourceforge.net).

For the whole-brain VBM analysis, the covariate of interest in our multiple regression model was risk tolerance as assessed by alpha. Age, gender and global GM volume (following analysis of covariance normalization) were included in the design matrix. F contrasts were applied first with $P < 0.001$ (uncorrected) as the criterion to detect voxels in which GMV significantly correlated with individual risk attitudes. Nonstationary whole-brain cluster-level correction[27] was then applied to correct for multiple comparisons at a threshold of $P < 0.05$.

**Ambiguous lotteries.** Pure risk choices were made as part of a larger study that included an additional 60 choices between a certain gain of $5 and ambiguous lotteries (20 amounts: $5–$120) whose exact probabilities were unknown (24, 50 and 74% of the chips were covered by an occluder). Consistent with previous results[4,7], we found no evidence that ambiguity aversion was linked to age and/or rPPC grey matter (see Supplementary Methods).

**Data availability.** All relevant data and analysis code are available from the authors upon reasonable request. rPPC ROI mask download available at https://yale.box.com/v/levylab-gilaie-dotan-etal-2014.

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

## Acknowledgements

Research supported by NIH grants R01 5R01AG033406 to P.W.G. and I.L, and R21AG049293 to I.L.

## Author contributions

M.A.G., A.T., S.G.-D., P.W.G. and I.L. designed research; M.A.G. performed research; M.A.G. analysed data with advice from the other authors; M.A.G, A.T., S.G.-D., P.W.G. and I.L wrote the paper.

## Additional information

**Competing financial interests:** The authors declare no competing financial interests.

**Publisher's note**: 

