## [Peer Review File · Nature Communications]

Reviewers' comments:

Reviewer #1 (Remarks to the Author):

The major finding of the manuscript is that a neurobiological marker of aging - the volume of gray matter in a discrete region of the right posterior parietal cortex- accounts for age-related changes in risk preference. Particularly interesting is the demonstration that is is not age per se, but the reduction of gray matter volume in rPPC that correlates with the changes in risk preference.

I find this result interesting and useful for different labs studying aging and trying to find and use neurobiological markers of aging.

Some more details on the data can be useful to a differentiated readership:

- try to better indicate the extent and location of the ROI shown only in Fig. 2A in two blurred and small images
- The description of data modeled in fig 2b, c, d and e is confused in the manuscript because authors describe how rPPC gray matter volume and risk tolerance vary as function of different parameters by applying various models, but only some of these are clearly shown in fig 2. Please improve the clarity in the result description.

Reviewer #2 (Remarks to the Author):

This manuscript reports an experiment linking risk aversion and brain structure (parietal cortex gray matter) across the lifespan in 52 adults aged 18-88. Prior studies from this group have established that parietal gray matter predicts risk aversion (Gilaie-Dotan et al 2014 J Neuro) and that risk aversion changes through the lifespan (Tymula et al 2013 PNAS). The present study elegantly brings these two themes together, showing that parietal gray matter also changes with age, and most importantly that this neurobiological marker is a better predictor of risk aversion than age itself. The paper is very nicely written and the authors do a great job of explaining why age related change in decision-making is such an important public health concern. The economic task and other methods are sound, and the economic modelling and statistical models are clearly explained. There is much to like here and the paper is broadly 'newsworthy'.

My views on the paper were dampened by a couple of points that suggest to me the authors are striving for a simple story. In the supplementary methods, two design features become clear: 1) the choice task was actually completed during an fMRI scan, and 2) the choice task involved a further 60 trials involving ambiguous decisions. The fMRI and ambiguity data do not feature here. Both are clearly relevant to their overarching aims, and the reader's interpretation of the data. Is ambiguity aversion also linked to age and/or parietal gray matter? Do the functional imaging responses in the parietal region have further predictive value over choice and structure? Clearly the authors will argue that these are important questions but for separate papers. The fMRI point bothers me more than the ambiguity data, as the scanner environment is unlikely to represent an emotionally neutral

baseline for the choice assessment - at the least, this should be clearly stated as a limitation in the main text.

The discussion to the paper side-steps an interesting finding in the Tymula et al paper that the age effect for risk aversion followed an inverted U. The relationship here (pg 4) is linear, which the authors say is 'predicted by the prior research'.

The parietal gray matter effect is detected in a data-driven ROI from the Gilaie-Dotan paper. This is fine, and a supplemental analysis shows that parietal gray matter is a better predictor than global gray matter volume. For imagers, the omission of the VBM whole-brain analysis will be quite glaring, particularly when the authors wish to conclude (pg 5) that the results are 'specific to rPPC gray matter decline' and that a 'neurobiologically comprehensive model best characterizes the choice process'. I could overlook such language if the authors did not have the voxel-based analysis in their immediate possession, but when they do, I think there is an obligation to at least show the reader those data.

Minor: The logic behind Model 3 reminded me of what social psychologists would call a mediation (also Figure 3). Could the authors add a statement about whether this is or is not a conventional 'mediator'.

Reviewer #3 (Remarks to the Author):

This study examined the relationship between right posterior parietal cortex (rPCC) and risk behavior in older individuals, with the background that this region has been previously associated with risk behavior in a younger cohort. The purported significance of such a study is that this type of knowledge will inform our ability to understand how decisions made by an aging population will impact political and economic processes. Using ROI analyses, authors report that decreased volume of the rPCC is associated with increased age and greater adversity of risk.

Findings from this study were not as robust as one might hope for given the narrow focus of the paper, and I question the import and validity of some of the statistics presented.

Overall, I am left not completely convinced that findings are regionally specific to the rPCC region. Several concerns are described as followed:

1. Is rPCC significantly related to risk adversity using a whole brain VBM analysis, controlling for age and gender? Findings showing significance using an ROI analysis are of interest, but not as robust as if this finding is additionally present using non-hypothesis driven whole brain VBM methodology. As the authors compare rPPC with total GM volume in subsequent analyses, the lack of whole brain VBM analysis seems like a significant omission.

2. Given potential gender differences in both brain volume and risk adversity, and unclear distribution of gender across the ages, all analyses should be adjusted for gender.

3. I question the validity of Models 4 and 5 somewhat, in that total GM volume includes the rPCC volume, and thus, when total GM volume and rPCC volume are in the same model together, rPCC is more heavily weighted than other brain regions. Does total GM volume alone, adjusted for age (and gender) predict risk adversity (without rPCC out in the model) and if so, how does this model compare with one that includes just rPCC adjusted for age and gender?

4. Some further discussion about the importance and relevance of these findings would enhance the manuscript.

5. Some further detail about the population should be included in the manuscript itself, and not just in the supplemental materials. MMSE does not screen for dementia, per se, which would require some type of functional assessment (i.e. activities of daily living, etc.). Is more accurate to say that MMSE was used to exclude overt cognitive impairment.

We thank the reviewers for their enthusiasm and for their helpful comments which have led to a substantially improved manuscript. Below, we address the major concerns.

Reviewer 1

Some more details on the data can be useful to a differentiated readership:

- try to better indicate the extent and location of the ROI shown only in Fig. 2A in two blurred and small images

-The description of data modeled in fig 2b, c, d and e is confused in the manuscript because authors describe how rPPC gray matter volume and risk tolerance vary as function of different parameters by applying various models, but only some of these are clearly shown in fig 2. Please improve the clarity in the result description.

Thank you for this – our description of the ROI was indeed lacking.

- To provide exact details of the ROI's location and extent we have added spatial extent and location coordinates to the main text, and we have indicated a URL from which a mask for this ROI can be downloaded and viewed or used for independent analyses (p. 4):

“Using voxel-based morphometry (VBM), we sampled gray matter volume in the rPPC region-of-interest which was defined independently based on an earlier study (Figure 2a; MNI coordinates 27, -78, 48; spatial extent, 1232 mm³; from Reference 7, Study 1; mask download available at <https://yale.box.com/v/levylab-gilaie-dotan-et-al-2014>).”

- To improve the clarity of the results, we modified the text in the results section to bring clarity to the findings and to better relate the results to the figures (p. 4-5).

Reviewer 2

My views on the paper were dampened by a couple of points that suggest to me the authors are striving for a simple story. In the supplementary methods, two design features become clear: 1) the choice task was actually completed during an fMRI scan, and 2) the choice task involved a further 60 trials involving ambiguous decisions. The fMRI and ambiguity data do not feature here. Both are clearly relevant to their overarching aims, and the reader's interpretation of the data. Is ambiguity aversion also linked to age and/or parietal gray matter? Do the functional imaging responses in the parietal region have further predictive value over choice and structure? Clearly the authors will argue that these are important questions but for separate papers. The fMRI point bothers me more than the ambiguity data, as the scanner environment is unlikely to represent an emotionally neutral baseline for the choice assessment - at the least, this should be clearly stated as a limitation in the main text.

Is ambiguity aversion also linked to age and/or parietal gray matter?

- This is an important question, which we have not addressed clearly. In this manuscript we focused on risk (and not ambiguity) attitudes, because we wanted to test a very specific hypothesis that was derived from two of our previously published manuscripts:
 - In Gilaie-Dotan et al 2014 J Neuro, the relationship between rPPC gray matter volume and risk tolerance was specific to risk. Ambiguous lotteries were included in Study 1 of that paper, but no significant relationship between ambiguity attitude and gray matter volume was found.

- In Tymula et al 2013 PNAS older adults were found to be more risk averse than young adults, but there was no difference in ambiguity attitude between young and older adults. Only the adolescent group in that study showed a significant difference in ambiguity attitudes, and this age group is not included in the current study.
- Taken together, the prediction from these studies is that ambiguity aversion is not linked to age in adulthood and/or to parietal gray matter.
- However, given the reviewer's question, we completed the following analyses:
 - We included the ambiguous trials and modeled the EU of each option using the same function employed in the two papers above. This function takes into account the effect of ambiguity on the perceived probability, and an ambiguity attitude parameter (β) can be estimated: $EU(v,p,A) = [p - \beta(A/2)] \times v^\alpha$
 - We allowed β , the ambiguity attitude parameter, to vary during the estimation procedure as a function of age (Model 1), as a function of age + rPPC gray matter volume (Model 2), and as a function of age + rPPC gray matter volume + global gray matter volume (Model 3).
 - We found no relationship between ambiguity aversion and any of these predictors (all p values > 0.2), in any of the models.
 - In a fourth model, we controlled for the relationship between risk attitude and rPPC gray matter volume by also allowing the risk attitude parameter to vary as a function of rPPC gray matter volume during the estimation procedure. Again, we found no relationship between ambiguity aversion and any of the predictors (age, rPPC gray matter volume, global gray matter volume; all p values > 0.2), and as expected, we found a significant positive relationship between risk attitude and rPPC gray matter volume ($z=3.56$, $p<0.001$).
- These methods and results are now presented in the Supplementary Information section.

The effect of the scanner environment on choice assessment:

- We agree with Reviewer 2 that the functional data will address important questions but in separate papers, and we acknowledge that, in theory, the scanner environment may not represent an emotionally neutral baseline for choice assessment. As requested, we have explicitly raised this possibility and pointed it out as a potential limitation in the main text, and we also present a comparison between age-specific risk attitudes estimated here and those obtained outside the scanner and presented in Tymula et al 2013, which may help to alleviate (though not eliminate) this concern.
- Using the results from Model 1 in our paper, we estimated what risk tolerance would be for a typical 21 year old and for a typical 90 year old. We then compared these estimates to those obtained in Tymula et al 2013 for young adults and older adults. We find that our estimates from the current study fall within the 95% confidence intervals for age-matched estimates obtained outside the scanner in Tymula et al 2013.

This is now described in p. 6-7:

“Choice data were collected in an MRI scanner during the acquisition of functional scans whose data are not presented here. Although in theory the scanner environment may affect individual risk attitudes, we note, that age-based estimates of risk tolerance derived from Model 1 are comparable to those obtained outside the scanner: the risk tolerance parameter (alpha) is predicted to drop slowly with each passing year, from 0.61 at 21 years of age to 0.42 by 90 years old. These estimates of

risk tolerance fall within the 95% confidence intervals for age-specific alpha values reported previously by our group⁴ in a task where choices were made on a desktop computer.”

The discussion to the paper side-steps an interesting finding in the Tymula et al paper that the age effect for risk aversion followed an inverted U. The relationship here (pg 4) is linear, which the authors say is 'predicted by the prior research'.

- This is a great point that we now explicitly address in the manuscript. The inverted-U relationship between risk attitude and age, observed in our previous study, was due to increased risk aversion in adolescents. Across the adult groups, a monotonic increase in risk aversion was observed. We have added the following paragraph to the Discussion (p. 7) to reiterate that the current sample of participants is comprised of young, midlife, and older adults only:

“The current study only includes adults over 18 years old. As in our previous study⁴, within this age range risk tolerance decreases monotonically. When adolescents are taken into account, however, the lifespan trajectory for risk tolerance may be described by a U-shaped function, with adolescents showing higher risk aversion compared to young and midlife adults. This raises an intriguing hypothesis for future research - that increased risk aversion in adolescence might be parsimoniously accounted for by changes in rPPC gray matter volume during childhood and adolescent development.”

Minor: The logic behind Model 3 reminded me of what social psychologists would call a mediation (also Figure 3). Could the authors add a statement about whether this is or is not a conventional 'mediator'.

- According to the definition of mediation provided by Baron and Kenny (1986), we do meet the requirements for conventional mediation. We show that there is a significant relationship between age and risk tolerance (Model 1, Table 1), that there is a significant relationship between age and rPPC gray matter volume (Figure 2B, stats in text), and that a) there is *no* significant relationship between age and risk tolerance when rPPC gray matter volume is included in the regression and b) there *is* a significant relationship between rPPC gray matter volume and risk tolerance when age is included in the regression (Model 3, Table 1).
- However, we are shying away from the term 'mediation' in the manuscript to avoid any semantic confusion among readers. We don't want to (as of yet) suggest a causal relationship between gray matter thinning and risk aversion. The relationship is still (at this point) correlational, it's just that the correlation is better accounted for by changing rPPC gray matter than by chronological age. This finding is an important precursor to direct tests of causality.
- If the reviewers and editor prefer that we state this explicitly in the manuscript, we will be happy to do so.

The parietal gray matter effect is detected in a data-driven ROI from the Gilaie-Dotan paper. This is fine, and a supplemental analysis shows that parietal gray matter is a better predictor than global gray matter volume. For imagers, the omission of the VBM whole-brain analysis will be quite glaring, particularly when the authors wish to conclude (pg 5) that the results are 'specific to rPPC gray matter decline' and that a

'neurobiologically comprehensive model best characterizes the choice process'. I could overlook such language if the authors did not have the voxel-based analysis in their immediate possession, but when they do, I think there is an obligation to at least show the reader those data.

- Thank you for this important comment. In the original manuscript we aimed to test a very specific hypothesis, based on our previous empirical findings. In Study 1 of Gilaie-Dotan et al 2014 J Neuro, exploratory whole-brain VBM analysis only revealed a single cluster, in the rPPC, whose volume predicted risk tolerance, a finding that was confirmed in Study 2 of the same paper. Furthermore, a whole brain analysis of Study 2 data, did not reveal any additional clusters that predicted risk tolerance. In the current study, we again confirm the robustness of the rPPC/risk tolerance relationship in an independent age-diverse sample of participants.
- Having said that, we agree with Reviewer 2 and Reviewer 3 (below) that additional whole brain VBM analysis to confirm that our results are specific to the rPPC in this third dataset is in order, so we now report the results of an exploratory whole brain analysis in the main text (p. 6). In line with both studies presented in Gilaie-Dotan et al 2014, no additional clusters were found to predict risk tolerance:

“While the primary aim of the current study was to test a specific hypothesis regarding the rPPC's role in modulating age-related changes in risk tolerance, we also conducted an exploratory whole brain VBM analysis to evaluate whether gray matter volume is predictive of risk tolerance in any additional brain regions. In a voxel-wise manner, multiple regression was used to compute the linear relationship between risk tolerance and gray matter volume, controlling for age, gender, and global gray matter volume. In line with the original findings upon which our primary hypothesis is based⁷, no additional clusters showed a significant relationship between gray matter volume and risk tolerance after the necessary correction for whole-brain multiple comparisons.”

Reviewer 3

Is rPCC significantly related to risk adversity using a whole brain VBM analysis, controlling for age and gender? Findings showing significance using an ROI analysis are of interest, but not as robust as if this finding is additionally present using non-hypothesis driven whole brain VBM methodology. As the authors compare rPPC with total GM volume in subsequent analyses, the lack of whole brain VBM analysis seems like a significant omission.

- Please see the response above to Reviewer 2

Given potential gender differences in both brain volume and risk adversity, and unclear distribution of gender across the ages, all analyses should be adjusted for gender.

- This is an important comment, and we apologize for neglecting to describe our gender analysis in the original manuscript. In a simple model where we allow risk tolerance to vary by gender during the estimation procedure (analogous to Model 1 but with a dummy predictor for gender instead of age), we found no effect of gender on risk tolerance ($Z=-0.78$, $p=0.44$), in line with the results from Tymula et al 2103, which also found no effect of gender on risk tolerance.

- We did not include gender in the 5 models for this reason, but this was a mistake on our part. Readers will undoubtedly be curious, so we now include the result of each model controlling for gender.
 - When gender is included in the 5 models, there is no effect of gender on risk tolerance in any model (all p-values > 0.46), and all other effects remain qualitatively unchanged.
 - We have added a table that includes gender for each of the five models to the supplementary section and have added the following to the main text:

“Controlling for gender in each model revealed no effect of gender on risk tolerance and did not qualitatively change the relationship between risk tolerance and age/rPPC gray matter (Table S1, Supplementary Information).”

I question the validity of Models 4 and 5 somewhat, in that total GM volume includes the rPCC volume, and thus, when total GM volume and rPCC volume are in the same model together, rPCC is more heavily weighted than other brain regions. Does total GM volume alone, adjusted for age (and gender) predict risk adversity (without rPCC out in the model) and if so, how does this model compare with one that includes just rPCC adjusted for age and gender?

- We apologize for not being clear on this. Models 4 and 5 were simply meant to serve as control analyses, with the main goal being to rule out the possibility that changes in risk tolerance could be explained by global, age-related changes in gray matter thickness, rather than being specific to local gray matter decline in the rPPC. As gray matter in most cortical areas declines with age, it was important to show that the effect of rPPC is robust to the inclusion of global gray matter volume in our model
- We have edited the topic sentence of this paragraph to make this goal more explicit in the main text (p. 5):

“Two additional models confirmed that these results are specific to local gray matter decline in the rPPC, rather than global, age-related changes in gray matter thickness.”
- In response to Reviewer 3’s questions:
 1. Does total GM volume alone, adjusted for age (and gender) predict risk adversity (without rPCC out in the model)?
 - with a strict $p < 0.05$ cutoff, no it does not. But the p-value is trending at 0.081 (more on this below).
 2. How does this model compare with one that includes just rPCC adjusted for age and gender?
 - see Model 3 in Supplementary Table 1 for full results, but in short, rPPC does predict risk tolerance in this model ($p=0.032$)
 - this model also has a lower BIC value than the “GM volume alone, adjusted for age and gender” model (3038 vs 3057), indicating that it better accounts for the data
 - Further, when we control for the relationship between rPPC and risk aversion, as well as age and gender, the p-value for global GM is no longer trending ($p=0.146$, Model 4, Table S1; $p=0.177$, Model 5, Table S1)

Some further discussion about the importance and relevance of these findings would enhance the manuscript.

- We have enhanced the discussion section to better convey the importance and relevance of our findings (p. 7-8).

Some further detail about the population should be included in the manuscript itself, and not just in the supplemental materials. MMSE does not screen for dementia, per se, which would require some type of functional assessment (i.e. activities of daily living, etc.). Is more accurate to say that MMSE was used to exclude overt cognitive impairment.

- We thank Reviewer 3 for improving the accuracy of our MMSE description and have adopted the language suggested (“used to exclude overt cognitive impairment” rather than “used to screen for dementia”).
- We have also incorporated more details about the population into the methods section of the main text rather than have them appear only in the Supplemental Information section

REVIEWERS' COMMENTS:

Reviewer #1 (Remarks to the Author):

The Authors clarified many of the unclear points.

Reviewer #2 (Remarks to the Author):

The authors are to be commended for a careful and rigorous job on the revision manuscript and response letter. The addition of the ambiguity aversion data in the supp, the comparison with their past non-fMRI data to evaluate the effect of the scanner environment, and the VBM analysis all help to instill a greater confidence in their findings, and convey better the richness of this important experiment. The only weak result in these new analysis is the whole brain VBM data, in which there are no significant voxels correlating with risk tolerance outside of parietal cortex (which is good), but it appears that the rPPC was also not statistically significant in the whole brain VBM. It is possible that the inclusion of multiple covariates and the stringent correction has served to wipe out all effects. This was the specific question raised by reviewer 3.

Reviewer #3 (Remarks to the Author):

The Authors of this manuscript did a thorough job of addressing my previous questions and concerns. I believe the revised version is novel, and adds to the current body of literature in this field.

We thank the reviewers for their final comments and for their enthusiasm regarding this manuscript. Below, we address one additional concern from Reviewer 2.

Reviewer 2

The authors are to be commended for a careful and rigorous job on the revision manuscript and response letter. The addition of the ambiguity aversion data in the supp, the comparison with their past non-fMRI data to evaluate the effect of the scanner environment, and the VBM analysis all help to instil a greater confidence in their findings, and convey better the richness of this important experiment. The only weak result in these new analysis is the whole brain VBM data, in which there are no significant voxels correlating with risk tolerance outside of parietal cortex (which is good), but it appears that the rPPC was also not statistically significant in the whole brain VBM. It is possible that the inclusion of multiple covariates and the stringent correction has served to wipe out all effects. This was the specific question raised by reviewer 3.

We very much appreciate Reviewer 2's feedback regarding the quality and rigor of the revision. As Reviewer 2 rightly points out, our rPPC findings were unable to surpass the stringent statistical corrections needed to control the false discovery rate in exploratory whole brain analyses. We have edited that paragraph in the manuscript to make this more explicit and have elaborated on its implications:

While the primary aim of the current study was to test a specific hypothesis regarding the rPPC's role in modulating age-related changes in risk tolerance, we also conducted an exploratory whole brain VBM analysis to evaluate whether gray matter volume is predictive of risk tolerance in any additional brain regions. In a voxel-wise manner, multiple regression was used to compute the linear relationship between risk tolerance and gray matter volume, controlling for age, gender, and global gray matter volume. No clusters showed a significant relationship between gray matter volume and risk tolerance after the stringent corrections needed to combat false discoveries in exploratory whole-brain analyses. Given that this is the third independent dataset showing a significant relationship between rPPC gray matter volume and risk tolerance, the likelihood that we are reporting a repeated false discovery is extremely low. While we cannot definitively rule out the possibility that additional regions' structure and function contribute to age-related changes in risk tolerance, our a priori hypothesis-driven ROI analyses point to a clear role of the rPPC in these processes.

We would like to stress that our ROI-based analyses were hypothesis-driven, derived directly from two previously published manuscripts. Additionally, the robustness of the rPPC-risk tolerance relationship was verified in an independent dataset in Gilaie-Dotan, S. *et al.* (2014). Here we again verify the replicability of that relationship in a third, age-diverse dataset. Whole brain correction needed to combat false positive results is clearly unnecessary in this case, but we appreciate that stringent corrections may also lead to the failed detection of real results in other brain regions. We hope that both of these points come through in the manuscript.